# Exercise Does Not Independently Improve Histological Outcomes in Biopsy-Proven Non-Alcoholic Fatty Liver Disease: A Systematic Review and Meta-Analysis

**DOI:** 10.3390/genes14091811

**Published:** 2023-09-17

**Authors:** George Chen, Bubu A. Banini, Albert Do, Craig Gunderson, Saif Zaman, Joseph K. Lim

**Affiliations:** 1Department of Internal Medicine, Yale School of Medicine, New Haven, CT 06510, USAsaif.zaman@yale.edu (S.Z.); 2Section of Digestive Diseases, Department of Internal Medicine, Yale School of Medicine, New Haven, CT 06510, USAjoseph.lim@yale.edu (J.K.L.); 3Department of Internal Medicine, VA Connecticut Health Care System, West Haven, CT 06516, USA

**Keywords:** non-alcoholic fatty liver disease, NAFLD, exercise, liver fibrosis, hepatic steatosis

## Abstract

Introduction: The independent effect of exercise on liver histology in non-alcoholic fatty liver disease (NAFLD) remains unclear. As such, we conducted a systematic review and meta-analysis of the effect of exercise alone on histological endpoints in biopsy-proven NAFLD. Materials and Methods: A systematic literature search was conducted to include controlled clinical trials investigating the effect of exercise alone on liver histology in biopsy-proven NAFLD. Meta-analysis was conducted for histological outcomes with available data from a minimum of three studies. Pooled estimates of the effect of exercise on histological endpoints were calculated using random-effects models. Results: We identified three controlled clinical trials that assessed the independent effect of exercise on histological outcomes in patients with biopsy-proven NAFLD. The studies consisted of 72 total participants, including 40 subjects in the exercise intervention and 32 individuals in the comparison group. Meta-analysis showed that exercise did not significantly improve Brunt grade, NAFLD activity score, and fibrosis in NAFLD. Discussion: Exercise alone may not lead to significant histopathological improvement in NAFLD. Future well-powered randomized controlled trials are needed to better characterize the impact of exercise on histological outcomes and clinical endpoints.

## 1. Introduction

Non-alcoholic fatty liver disease (NAFLD) is the most common etiology of chronic liver disease worldwide [1]. Weight and fat mass reduction achieved via a combination of behavioral exercise and dietary modification remain the mainstay of treatment for NAFLD [2]. Although previous randomized controlled trials (RCTs) and meta-analyses have demonstrated that weight reduction improves histological outcomes in NAFLD in a dose–response relationship [2,3,4], the effect of exercise alone on liver histology remains unclear. To this end, we conducted a systematic review and meta-analysis of the independent effect of exercise on histological outcomes in biopsy-proven NAFLD.

## 2. Materials and Methods

A systematic search of the literature using Embase, PubMed, and Web of Science databases from inception to October 2022 was conducted in accordance with PRISMA guidelines (Figure 1). Articles were chosen using the following inclusion criteria: (1) involved human participants with biopsy-proven NAFLD, (2) studied the effect of exercise alone on histological endpoints, and (3) were controlled clinical trials. Histological outcomes included hepatic steatosis, hepatocyte ballooning, lobular inflammation, Brunt grade, NAFLD activity score (NAS), and liver fibrosis. Meta-analysis was performed for each histological outcome with a mean difference between pre- and post-intervention results and the standard deviation of mean difference available from a minimum of three studies. The pooled effect size of exercise on histological outcomes was estimated by calculating the standardized mean difference in random-effects models. Statistical heterogeneity between studies was quantified using the I^2^ index.

## 3. Results

### 3.1. Study Selection

Our initial literature search yielded 461 articles. After removing duplicates and studies not meeting the inclusion criteria, two articles remained. One additional study was selected from a manual search of reference lists of retrieved manuscripts. In sum, we identified three controlled clinical trials that assessed the independent effect of exercise on liver histology in patients with biopsy-proven NAFLD (Figure 1) [5,6,7].

### 3.2. Participant and Study Characteristics

The three studies comprised a total of 72 participants (approximately 49% female), including 40 individuals in the exercise group and 32 subjects in the comparison group. Participant characteristics are summarized in Table 1. The mean age ranged from 48 to 60 years, with a mean body mass index of 34 to 35. Subjects with alternative etiologies of liver disease were excluded in all studies [5,6,7].

Study intervention protocols are described in Table 2. Hickman et al. conducted an RCT comparing 24 weeks of moderate-intensity resistance exercise with a hypocaloric diet. The exercise group participated in a circuit exercise program consisting of 15 resistance exercises on pneumatic strength training machines targeting main muscle groups. Supervised training was conducted in 30 s exercise periods followed by 30 s rest intervals. Subjects underwent three sessions per week with gradually increased session durations from 12 to 60 min by the fifth week. Participants in the diet group followed a hypocaloric diet aimed to induce 5% to 10% weight loss in 16 weeks without changing physical activity levels. The amount of calorie restriction was individually calculated for each participant based on their predicted energy needs and expenditures. In addition, participants were advised to follow a diet consisting of 40% fat, 40% carbohydrate, and 20% protein while decreasing saturated fat and sugar intake. Two participants in the exercise group and two participants in the diet group were unable to complete the study. Paired liver biopsies were performed at the beginning of the intervention and 24 weeks [5].

In another RCT, Eckard et al. compared 24 weeks of moderate-intensity combined aerobic and resistance exercise with a standard-of-care control group. The exercise regimen involved treadmills, stationary bikes, and arm and leg strength exercises in a supervised setting. The exercise intervention comprised four to seven sessions each week with session lengths of 20–60 min. Details regarding the standard-of-care control group were not provided. Six participants in the exercise group and three participants in the standard of care group withdrew from the study. Paired liver biopsies were performed at the start of the study and 24 weeks [6].

Naimimohasses et al. conducted a non-RCT comparing 12 weeks of moderate-intensity aerobic exercise with a moderately hypocaloric diet. Participants in the exercise group performed aerobic exercise using treadmills, elliptical trainers, or stationary bikes. The exercise sessions gradually increased in frequency from three to five sessions per week and duration from 21 to 42 min over 12 weeks. Two sessions per week were supervised, while the remaining one to three sessions were unsupervised. Participants in the hypocaloric diet group were encouraged to follow a Mediterranean-like diet and were provided with recipes focused on whole foods, legumes, vegetables, nuts, fish, and complex carbohydrates while minimizing meats, saturated fats, and processed foods without changing physical activity levels. Six participants in the exercise group and two participants in the diet group were unable to complete the study. Paired liver biopsies were conducted at the start of the interventions and week 13 [7].

### 3.3. Effect of Exercise on Histological Outcomes

The effect of exercise and comparison groups on histological outcomes in each study are summarized in Table 2. Hickman et al. reported no significant histological improvement following exercise intervention, while the diet group experienced a reduction in steatosis (*p* = 0.04) and NAS (*p* = 0.05) [5]. In the study by Eckard et al., neither the exercise nor control group achieved significant improvement in histological endpoints, although only Brunt grade, NAS, and fibrosis were reported [6]. Naimimohasses et al. demonstrated exercise-associated improvements in hepatocyte ballooning (*p* = 0.02) and fibrosis (*p* = 0.04), while the diet group had significant improvement in steatosis (*p* = 0.004) and NAS (*p* = 0.01) [7].

Among measured histological outcomes, Brunt grade, NAS, and liver fibrosis met the criteria for meta-analysis with data available from a minimum of three studies. The pooled effect size of exercise on Brunt grade (mean difference [MD] −0.01; 95% confidence interval [CI] −0.48 to 0.45; *p* = 0.96), NAS (MD 0.61; 95% CI −0.44 to 1.66; *p* = 0.26), and liver fibrosis (MD −0.18; 95% CI −0.95 to 0.60; *p* = 0.65) was not statistically significant compared to the comparison groups (Figure 2, Figure 3 and Figure 4). High heterogeneity was observed between the studies for all three histological outcomes (I^2^ = 75% for Brunt grade; 71% for NAS; 75% for liver fibrosis).

## 4. Discussion

In this meta-analysis, the first to address this question to our knowledge, we found that exercise did not independently affect histological outcomes in patients with biopsy-proven NAFLD. The exercise protocols implemented in these studies were of moderate intensity, ranged from 12 to 24 weeks, consisted of 20 to 60 min sessions three to seven times per week, and included aerobic exercise, resistance training, or a combination of both. Although previous RCTs have shown that exercise independently improves serum and imaging-based markers of hepatic steatosis and fibrosis [8,9], our findings suggest that exercise alone does not significantly impact Brunt grade, NAS, or liver fibrosis. Exercise has been shown to increase insulin sensitivity, inhibit inflammatory pathways, and reduce fibrogenesis in animal models [10,11,12], but it is plausible that these effects may not directly translate to significant histopathological or clinical improvement in patients with NAFLD without additional weight reduction or dietary modification.

We acknowledge multiple limitations of our meta-analysis, including the relatively small sample sizes with high outcome heterogeneity resulting in underpowered analysis and the use of different comparison groups across studies. In addition, our findings may not be extrapolated to lean patients with NAFLD due to the enrichment of subjects with BMI > 30 in these studies. Additional well-powered RCTs are needed to further investigate the impact of exercise on histological endpoints, including a reduction in hepatic steatosis, NASH resolution, and fibrosis improvement, as well as liver and cardiometabolic outcomes.

## Figures and Tables

**Figure 1 genes-14-01811-f001:**
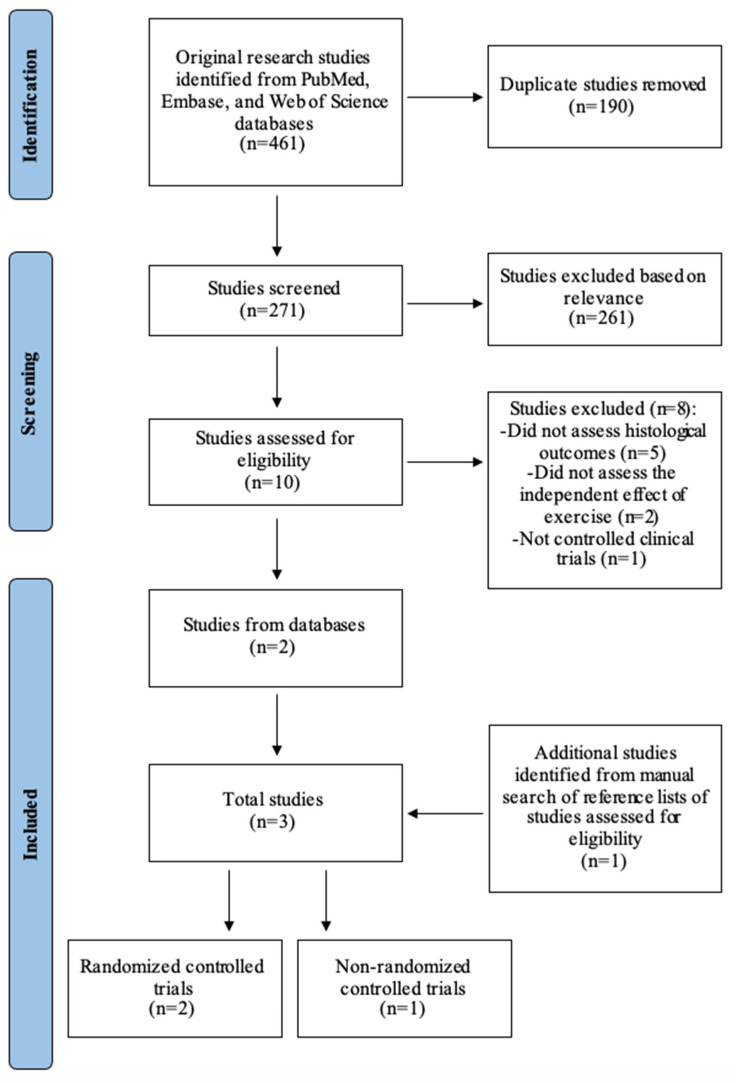
Selection of studies for review and meta-analysis.

**Figure 2 genes-14-01811-f002:**
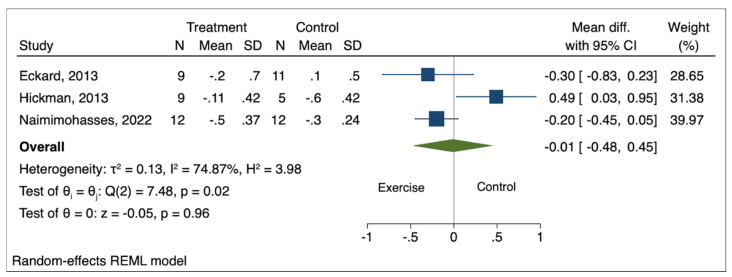
Pooled effect of exercise on Brunt grade in biopsy-proven NAFLD [5,6,7].

**Figure 3 genes-14-01811-f003:**
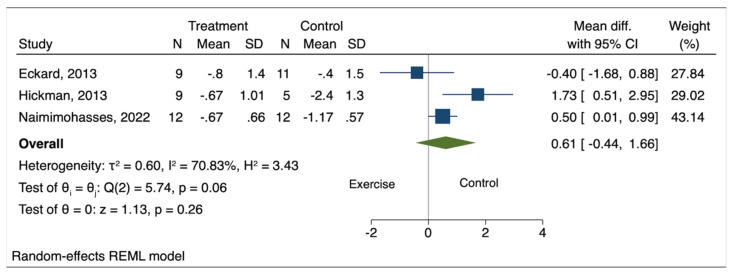
Pooled effect of exercise on NAFLD activity score in biopsy-proven NAFLD [5,6,7].

**Figure 4 genes-14-01811-f004:**
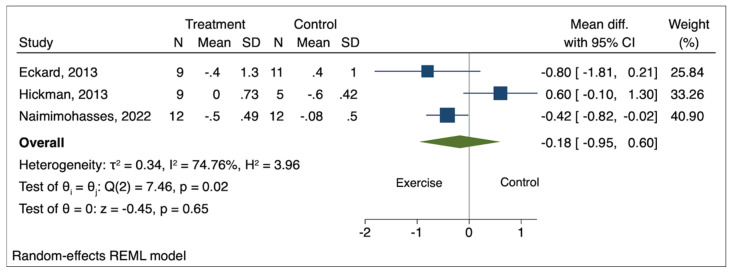
Pooled effect of exercise on liver fibrosis in biopsy-proven NAFLD [5,6,7].

**Table 1 genes-14-01811-t001:** Study participant characteristics.

	Hickman et al.	Eckard et al.	Naimimohasses et al.
Study design	RCT	RCT	Non-RCT
Mean age (years)	48	51	60
Sex (% female)	38	35	65
BMI (kg/m^2^)	34	34	35
Diabetes (%)	0	20	61
Hypertension (%)	NR	80	55
Metabolic syndrome (%)	62	NR	52
Total sample size (*n*)	21	20	31
Exercise group (*n*)	13	11	16
Comparison group (*n*)	8	9	15

Abbreviations: BMI = body mass index; kg = kilogram; m = meter; *n* = number; RCT = randomized controlled trial; NR = not reported.

**Table 2 genes-14-01811-t002:** Study intervention protocols.

			Exercise Protocol				
Study Number	Exercise and Control Group Interventions	Mean Change in Weight and BMI	Intervention Duration (Weeks)	Frequency (Sessions per Week)	Session Duration	Intensity	Histological Outcomes
1	EG: circuit-based resistance exercise (n = 13) DG: hypocaloric diet (n = 8)	Weight: EG: “Statistically insignificant” (exact change NR) DG: −9.7% BMI: EG: 0 kg/m^2^ DG: −3 kg/m^2^	24	3	Initially 1 circuit (12 min), gradually increased to 5 circuits (60 min) by week 5	Moderate (50% of 1-RM)	EG: Steatosis: Unchanged (*p* = 0.12) Lobular inflammation: Unchanged (*p* = 0.77) Hepatocyte ballooning: Unchanged (*p* = 0.34) Brunt grade: Individual data reported but change not analyzed NAS: Unchanged (*p* = 0.29) Liver fibrosis: Unchanged (*p* = 1.0) DG: **Steatosis: Improved (*p* = 0.04)** Lobular inflammation: Unchanged (*p* = 0.17) Hepatocyte ballooning: Unchanged (*p* = 0.50) Brunt grade: Individual data reported but change not analyzed **NAS: Improved (*p* = 0.05)** Fibrosis: Unchanged (*p* = 0.50)
2	EG: aerobic and resistance exercise (n = 9) CG: No intervention (n = 11)	Weight: EG: +0.1 lb (95% CI −3.6 to +0.6) CG: −2.5 lb (−6.0 to 1.0) BMI: EG: NR CG: NR	24	4–7	20–60 min	Moderate	EG: Steatosis: Not provided Lobular inflammation: Not provided Hepatocyte ballooning: Not provided Brunt grade: Not changed (MD −0.2; 95% CI −0.7 to +0.3) NAS: Unchanged (MD −0.8; −1.8 to +0.3) Fibrosis: Unchanged (MD −0.4; −1.5 to +0.6) CG: Steatosis: Not provided Lobular inflammation: Not provided Hepatocyte ballooning: Not provided Brunt grade: Unchanged (MD −0.4; −1.4 to +0.6) NAS: Unchanged (MD −0.4; −1.4 to +0.6) Fibrosis: Unchanged (MD +0.4; −0.3 to +1.1)
3	EG: aerobic exercise (n = 16) DG: moderately hypocaloric diet (n = 15)	Weight: EG: −2 kg (*p* = 0.0005) DG: −7 kg (*p* < 0.0001) BMI: EG: −1.1 kg/m^2^ (*p* < 0.0001) DG: −1.9 kg/m^2^ (*p* = 0.0002)	12	3–5	Initially 21 min, gradually increased to 42 min over 12 weeks + 5–7 min warm-up and cool-down	Moderate (40–75% HR reserve)	EG: Steatosis: Unchanged (*p* = 0.50) Lobular inflammation: Unchanged (*p* = 0.50) **Hepatocyte ballooning: Improved (*p* = 0.02)** Brunt grade: Unchanged (*p* = 0.14) NAS: Unchanged (*p* = 0.09) Fibrosis: Improved (*p* = 0.04) DG: **Steatosis: Improved (*p* = 0.004)** Lobular inflammation: Unchanged (*p* = 0.38) Hepatocyte ballooning: Unchanged (*p* = 0.11) Brunt grade: Unchanged (*p* = 0.16) **NAS: Improved (*p* = 0.01)** Fibrosis: Unchanged (*p* = 0.50)

Abbreviations: BMI = body mass index; EG = exercise group; n = number; DG = diet group; NR = not reported; kg = kilogram; m = meter; RM = repetition maximum; NAS = Non-alcoholic fatty liver disease activity score; CG = control group; lb = pound; MD = mean difference; CI = confidence interval; HR = heart rate.

## Data Availability

No additional data were created.

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
