# Peer review of "Exercise Does Not Independently Improve Histological Outcomes in Biopsy-Proven Non-Alcoholic Fatty Liver Disease: A Systematic Review and Meta-Analysis"

_genes, 2023, doi:10.3390/genes14091811_

Round 1
Reviewer 1 Report
The study seems fine. I see one main issues:
- The authors need to state clearly that the available data are underpowered to answer the question
Reviewer 2 Report
General comments
Chen et al. conducted a systemic review and meta-analysis of the impact of exercise on liver histology change in NAFLD. They included three controlled trials and found that exercise without weight loss may not lead to significant histopathological improvement in NAFLD. Generally, this is an interesting and clinically relevant topic that provides new information in this field. There are a few comments listed below.
1. In the discussion, the author concluded exercise without significant weight reduction does not significantly impact Brunt grade, NAS, or liver fibrosis. However, I found no analysis for the weight reduction and the histology outcomes. Please clarify it.
2. The layout of Table 1 makes it difficult to compare the characteristics between individual studies. The authors may try to change the table layout to let individual studies in each column while the descriptions or outcomes of the studies are in rows.
3. Because this study mainly focused on exercise effects on NAFLD, I suggest the authors compare the exercise protocol in these 3 studies in more detail.
4. The control group comprises a hypocaloric diet, a standard-of-care diet, and a moderate hypocaloric diet. Detailed information on these diet regimens should be provided and compared. Do patients in the control group also receive exercise?
5. Participants' exercise or diet control protocol compliance should be mentioned. Do any participants withdraw from individual studies?
6. Because patients received paired biopsies before and after the intervention. What is the timing of liver biopsy in each study? What are the comorbidities of participants of individual studies?
7. The authors should report and discuss any publication bias from the selected trials.
Reviewer 3 Report
The authors discuss a systematic review and meta-analysis of the independent effect of exercise on histological outcomes in biopsy-proven non-alcoholic fatty liver disease (NAFLD). The study selected three controlled clinical trials that assessed the impact of exercise on liver histology in patients with NAFLD. The findings suggest that exercise alone may not independently improve histological outcomes in NAFLD without additional weight reduction or dietary modification. The authors highlight the limitations of the meta-analysis that includes the small number of eligible studies and the use of different comparison groups. Further well-powered randomized controlled trials are needed to investigate the impact of exercise on histological endpoints and liver and cardiometabolic outcomes in NAFLD. The authors presented the data in clear and concise manner and I believe the paper makes a significant contribution to the field and recommend the article to be published in its original form.
